# The Phytochemical Synergistic Properties of Combination of Bergamot Polyphenolic Fraction and *Cynara cardunculus* Extract in Non-Alcoholic Fatty Liver Disease

Jessica Maiuolo [1,*], Rocco Mollace [2], Francesca Bosco [3], Federica Scarano [3], Francesca Oppedisano [3], Saverio Nucera [3], Stefano Ruga [3], Lorenza Guarnieri [3], Roberta Macri [3], Irene Bava [3], Cristina Carresi [3], Micaela Gliozzi [3], Vincenzo Musolino [1], Antonio Cardamone [3], Anna Rita Coppoletta [3], Andrea Barillaro [3], Virginia Simari [3], Daniela Salvemini [4], Ernesto Palma [3,†] and Vincenzo Mollace [3,5,*,†]

1   Laboratory of Pharmaceutical Biology, IRC-FSH Center, Department of Health Sciences, University "Magna Græcia" of Catanzaro, 88100 Catanzaro, Italy
2   San Raffaele Telematic University, 00163 Rome, Italy
3   IRC-FSH Center Department of Health Sciences, University "Magna Græcia" of Catanzaro, 88100 Catanzaro, Italy
4   University of St. Luois, St. Louis, MO 63103, USA
5   Nutramed S.c.a.r.l, Roccelletta di Borgia, 88021 Catanzaro, Italy
*   Correspondence: maiuolo@unicz.it (J.M.); mollace@libero.it (V.M.)
†   These authors contributed equally to this work.

**Abstract:** Non-alcoholic fatty liver disease (NAFLD) is considered one of the leading causes of liver-related morbidity and mortality. NAFLD is a cluster of liver disorders that includes the accumulation of fat in the liver, insulin resistance, diffuse steatosis, lobular inflammation, fibrosis, cirrhosis and, in the latter stages, liver cancer. Due to the complexity of the disease and the multifactorial basis for the development of liver dysfunction, there is currently no unique drug treatment for NAFLD and the pharmacological options are inconclusive. In recent years, natural products have been studied for their potential beneficial effect in both preventing and treating fatty liver and its consequences in both local and systemic effects related to NAFLD. In particular, bergamot polyphenolic fraction (BPF), which is rich in natural polyphenols, and *Cynara cardunculus* wild type (which contains large quantities of sesquiterpenes, caffeic acid derivatives and luteolin) have both been investigated in both pre-clinical settings and clinical studies showing their effect in counteracting NAFLD-related health issues. In the present review we summarize the experimental and clinical evidence on the effect of BPF and Cynara extract alone or in their combination product (Bergacyn®) in NAFLD. In particular, data reported show that both extracts may synergize in counteracting the pathophysiological basis of NAFLD by inhibiting lipid accumulation in liver cells, oxidative stress and inflammation subsequent to liver syeatosis and, in the latter stages, liver fibrosis and tissue degeneration. Moreover, due to its powerful vasoprotective effect, the combination of BPF and Cynara extract (Bergacyn®) leads to improved endothelial dysfunction and cardioprotective response in both animal models of NAFLD, in veterinary medicine and in humans. Thus, supplementation with BPF and *Cynara cardunculus* extract and their combination product (Bergacyn®) represent a novel and potentially useful approach in preventing and treating NAFLD-associated complications.

**Keywords:** bergamot; bergamot polyphenolic fraction (BPF); *Cynara cardunculus* extract; non-alcoholic fatty liver disease (NAFLD); lipid lowering effects; lipid lowering nutraceuticals

## 1. Introduction

Non-alcoholic fatty liver disease (NAFLD) is considered one of the leading causes of liver-related mortality and morbidity and, to date, no approved therapies exist. NAFLD is a cluster of liver disorders that can evolve over time in a multi-step process: the first

stage leads to an increase in liver fat and involvement of inflammation. In fact, the initial feature of NAFLD manifestation is the accumulation of fats in the liver and the induced insulin resistance [1,2]. An initially benign condition, it can turn into severe non-alcoholic steatohepatitis, with diffuse steatosis, robust hepatocellular alteration, lobular inflammation, fibrosis and cirrhosis. Finally, NAFLD can degenerate to a terminal stage that involves death due to the onset of liver cancer or requires organ transplantation [3,4]. Today, NAFLD is considered a very common chronic liver disease that affects about 25% of the adult population representing the highest liver-related mortality worldwide and which, for this reason, constitutes a significant economic burden [5]. Recently, it has been shown that NAFLD has a close association with metabolic syndrome: in fact, NAFLD is stated in 47.3–63.7% of people with type 2 diabetes and in 80% of people with obesity [6]. In addition, both adults and children with fatty liver have shown changes in glucose and lipid metabolism [7]. Although NAFLD has also been found in lean subjects, obesity is a key factor in the development of this pathology and most patients are overweight, obese or characterized by high BMI. For this reason, NAFLD may be a predictor of metabolic syndrome, but data in the literature do not unequivocally clarify what is the cause and what the consequence [8]. The main cause of NAFLD is overnutrition, responsible for increasing fat deposits, initially in adipose tissue and subsequently, when this body system is saturated and dysfunctional, outside the anatomical seats designated for fat storage, generating ectopic fat. At the same time, macrophage infiltration of visceral adipose tissue occurs, which creates a pro-inflammatory state. At a later stage, inadequate lipolysis occurs which causes accumulation of liver fat. The alteration of lipid metabolism leads to the formation of lipotoxic lipids, responsible for cellular oxidative stress, dysfunction of cellular organelles (such as mitochondria and endoplasmic reticulum), activation of inflammation, fibrogenesis and apoptotic cell death [9]. Nevertheless, it is important to note that NAFLD can also be generated by other causes. For example, there is a recognized hereditary component and, although it is not responsible for a high increase in the risk of disease, the single nucleotide polymorphism in the PNPLA3 gene is the best characterized genetic variant associated with the occurrence of NAFLD [10,11]. A correct diagnosis of NAFLD requires assessment of hepatic steatosis by imaging techniques, although liver biopsy remains the gold standard for this diagnosis and to differentiate the severe condition from simple steatosis. Since it is not ethically advisable to perform liver biopsy in all suspected patients, some non-invasive imaging techniques are chosen to identify fatty liver infiltration, including computed tomography (CT), ultrasound (US), magnetic resonance imaging (MRI), and proton magnetic resonance spectroscopy (MRS) [12,13]. A representation of the evolution of NAFLD, is shown in Figure 1. Due to the complexity of the disease and the different stages, there is currently no unique drug treatment for NAFLD and pharmacological options were inconclusive [14]. Lifestyle-based interventions, which include physical exercise, weight loss and a qualitatively and quantitatively balanced diet, are considered essential tools for NAFLD management [15]. In particular, body weight loss, through a low caloric diet, has been associated with the improvement of NAFLD lesions, appreciated by means of histological validation, the reduction of liver fat, fibrosis and inflammation [16]. However, it is appropriate to note that an unbalanced diet (with low carbohydrate intake-excess fat) or extreme and rapid weight loss can have the opposite effects and aggravate the disease, inducing insulin resistance [17]. These findings are important to demonstrate that NAFLD patients should be treated with balanced diets not only quantitatively but also qualitatively: the diet should be characterized by the intake of both macronutrients and micronutrients, have reduced intake of saturated fats and fructose, have higher intake of lean proteins, fibers and polyunsaturated fatty acids [18]. Therefore, in light of what has been said, the purpose of this review is to understand if the dietary intake of certain plant extracts can contribute to the cure or slowdown of NAFLD. In particular, the effects of the polyphenolic fraction of bergamot, Cynara Cardunculus extract and a combination of these two products (Bergacyn®), will be investigated.

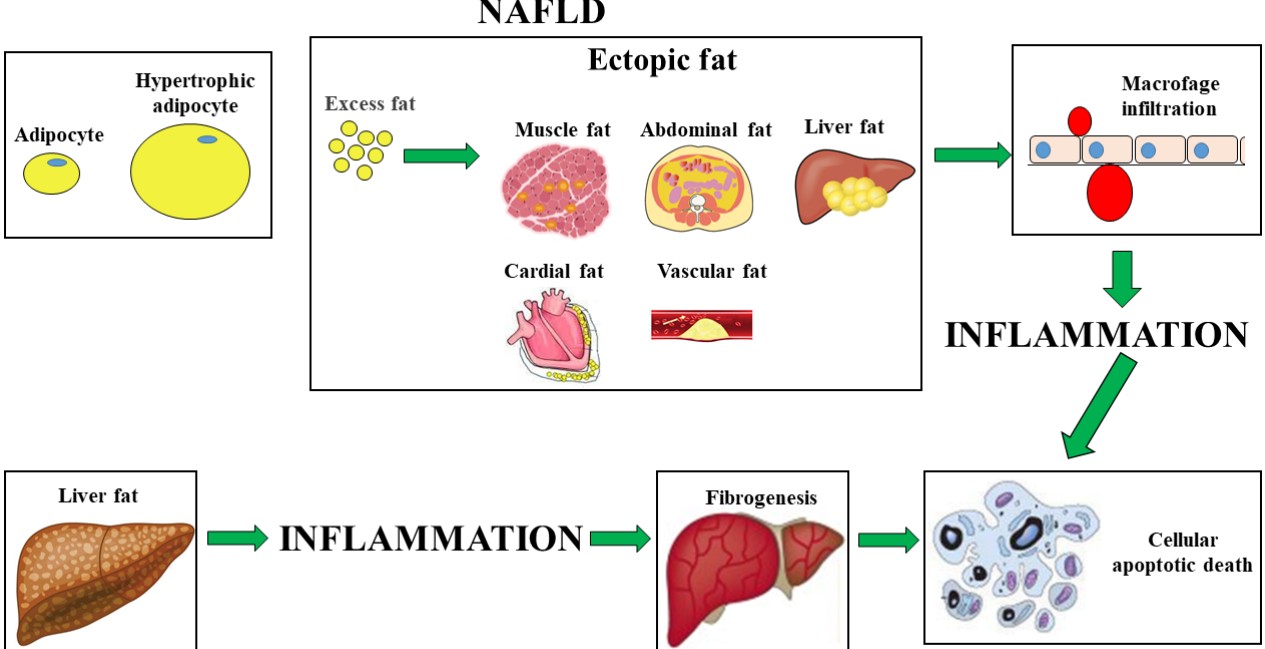

**Figure 1.** Representation of the evolution of NAFLD. The first box shows the hypertrophy of adipocyte that leads to an increase in cell size. The second box highlights the formation of ectopic fat, resulting from excessive fat gain, which involves other organs such as the liver, muscles, abdomen, heart and vascular system. The presence of ectopic fat promotes the infiltration of blood lymphocytes (the third box) and the onset of inflammation, responsible for apoptotic cell death (fourth box). Additionally, the fatty liver, if not treated, causes inflammation, which can evolve into fibrogenesis (second box from the left) and, in the final stage, into cellular apoptosis (third box from the left).

*Nutritional Interventions in NAFLD*

The American Association for the Study of Liver Diseases and The European Association for the Study of the Liver recommend a reduction in body weight by about 7–10% as a principal target in the treatment of NAFLD, overweight and obesity [19]. Several studies have shown that typical diets of the Western World are related to the development of metabolic diseases, including NAFLD [20,21], and characterised by a high consumption of refined cereals, red meat, sugar and a concomitant low intake of cereals, whole grains, vegetables and fruits. In fact, Western diets are described as "total high energy" with macronutrients that affect the metabolic pathways responsible for the accumulation of fat in the liver and the stimulation of lipogenesis [22,23]. The intake of specific nutraceuticals, which limit or reduce the effects of NAFLD, has been aimed at the description of two large families: those that have antioxidant and anti-inflammatory properties. Oxidative stress occurs when the balance between the production of reactive oxygen species (ROS) and the antioxidant physiological capacity is altered. High concentrations of ROS cause oxidative changes to cellular biological macromolecules. NAFLD is characterized by increased hepatic lipogenesis and altered degradation of free fatty acids, which are responsible for the accumulation of reactive species and the onset of oxidative stress. ROS and the accumulation of damaged macromolecules produce liver lesions [24] and trigger unregulated oxidative signals [25]. Therefore, oxidative stress is one of the main factors contributing to the progression from simple steatosis to steatohepatitis and several studies have described an antioxidant status in NAFLD [26]. Since ROS in not excessive amounts are necessary for physiological processes, a therapeutic strategy in NAFLD could be to reduce oxidative stress without seriously altering the general redox homeostasis [27]. Lipotoxicity, recognized in NAFLD as a result of lipid overload, causes stress signals to hepatocytes that trigger the activation of inflammatory pathways. When inflammation continues over an extended period, it can cause chronic injury, fibrogenesis and cell death [28]. For this reason,

the protective role of some nutraceuticals against inflammation in NAFLD should be examined. The available results have shown that pharmacological options for the treatment of NAFLD are inconclusive [29] and that the best treatment is to maintain an optimal lifestyle that includes exercise and a balanced diet. In fact, a reduction in body weight of 7–10% results in histological improvement and reduction of liver fat, fibrosis and necroinflammation [30]. To date, NAFLD treatment consists of limiting the intake of calories, fat (trans fatty acids, saturated fatty acids), fructose and, at the same time, increasing the intake of lean protein, fiber and PUFAs [31]. The Mediterranean diet is a nutritional model born and used by the populations living in the areas surrounding the Mediterranean Sea, based on the consumption of unrefined cereals, vegetables and fresh fruits, olive oil and nuts. In addition, the intake of fish, white meat, legumes and red wine must be carried out in moderation. Finally, the consumption of red meat, processed meat and sweets must be limited. This dietary model results, therefore, in a high content of complex carbohydrates and fibers, a low consumption of saturated fats and cholesterol and a high consumption of monounsaturated fatty acid (MUFA), with a balanced ratio polyunsatures fatty acids (PUFA) omega-6 to omega-3 [32]. It has been shown that the Mediterranean diet is able to reduce the risk of development and progression of NAFLD thanks to the antioxidant and anti-inflammatory effect of many bioactive nutraceutical and phytochemical compounds, contained in the main recommended foods. In addition, these effects can also benefit the production of metabolites associated with the intestinal microbiota [33–35]. Aller et al. have shown, in a study that involved 82 adult subjects with full-blown NAFLD, adherence to the Mediterranean dietary model reduced the histological characteristics of severe steatosis and insulin resistance [36,37]. Another study involving 1199 overweight adults found that NAFLD patients did not fully adhere to the Mediterranean diet [38]. At the same time, Baratta et al. showed that the Mediterranean diet was inversely related to NAFLD and that reduced not only the histological characteristics of NAFLD, but also the cardio-metabolic risk [39]. The protective and curative role of the Mediterranean diet in NAFLD patients was also highlighted by a study in which patients, divided into three groups, were each subjected to a type of diet with equal energy intake: the first group followed the diet against diabetes, as recommended by the American Diabetes Association; the second group joined a diet with a low glycemic index; finally, the third group adhered to a Mediterranean diet. The results showed that only patients on the Mediterranean diet had lower enzyme alanine amino-transferase values, showing lower liver suffering [40]. Misciagna et al. administered a control diet (based on the guidelines of the National Research Institute) or a low glycemic index Mediterranean diet for patients, divided into two groups, with ultrasound diagnosis of moderate/severe NAFLD. Only the Mediterranean diet with low glycemic index determined the reduction of liver fat [41]. Finally, an interesting study was conducted on patients who were normal weight ($n = 11$) or obese ($n = 35$) with NAFLD, to whom the Mediterranean diet was administered. At the end of the study, metabolic parameters and risk indices for cardiovascular disease were measured. The results showed a reduction in the severity of hepatic steatosis from 93% to 48%, and a meaningful reduction of the metabolites parameters and the levels of hepatic enzymes [42].

The mechanisms through which the Mediterranean diet is able to reduce the severity of NAFLD can be summed up in three key points: (1) the antioxidant effect; (2) the anti-inflammatory effect; (3) the effect on the gut microbiota [29]. Since inflammation and oxidative stress play a central role in the pathogenesis of NAFLD, the protective effects of most of the compounds found in foods of the Mediterranean diet are of critical importance. In particular, vitamins C, D and E reduce cellular stress, decrease levels of mitochondrial reactive oxygen species generation, increase the levels of antioxidant enzymes and prevent NAFLD progression by immunomodulatory, anti-inflammatory and anti-fibrotic properties [43,44]. Carotenoids, a class of natural fat-soluble pigments contained in different fruits and vegetables, also act as protective agents in NAFLD, demonstrating strong antioxidative effects and reducing inflammation and steatosis [45]. Polyphenols, a heterogenic group of bioactive compounds characterized by a phenolic structure, are present in all typical

foods of the Mediterranean diet (whole grains, fresh fruit, vegetables, olive oil, nuts, red wine) and demonstrate antioxidant and anti-inflammatory activities [46,47]. The liver is closely connected to the intestine, receiving 70% of the blood that flows into it. The existing crosstalk between these organs has generated a real "gut-liver axis" and, for this reason, intestinal dysfunction is related to many liver diseases, including NAFLD. The mechanisms of the gut microbiota involved in NAFLD include increased metabolites (such as short chain fatty acids, bile acids, lipopolysaccharides), dysfunction of the intestinal barrier, predisposition to obesity, induction of insulin resistance and liver inflammation [48,49]. A balanced diet model, such as the Mediterranean diet, can prevent or reduce intestinal dysfunction, affecting the diversity and composition of the microbiota. In particular, it has been shown that the Mediterranean diet promoted beneficial changes in the gut microbiota, reducing the *Firmicutes* and increasing the *Bacteroides* and that these alterations ameliorate inflammation, the metabolic alterations and obesity [50]. Polyphenols, contained in the Mediterranean diet, induce an increase in *Bifidobacteria*, that is associated with many metabolic benefits including a decrease of C-reactive protein, a plasma cholesterol reduction and lipid-lowering effect [51]. In Figure 2, the mechanisms through which the Mediterranean diet protects NAFLD patients are represented.

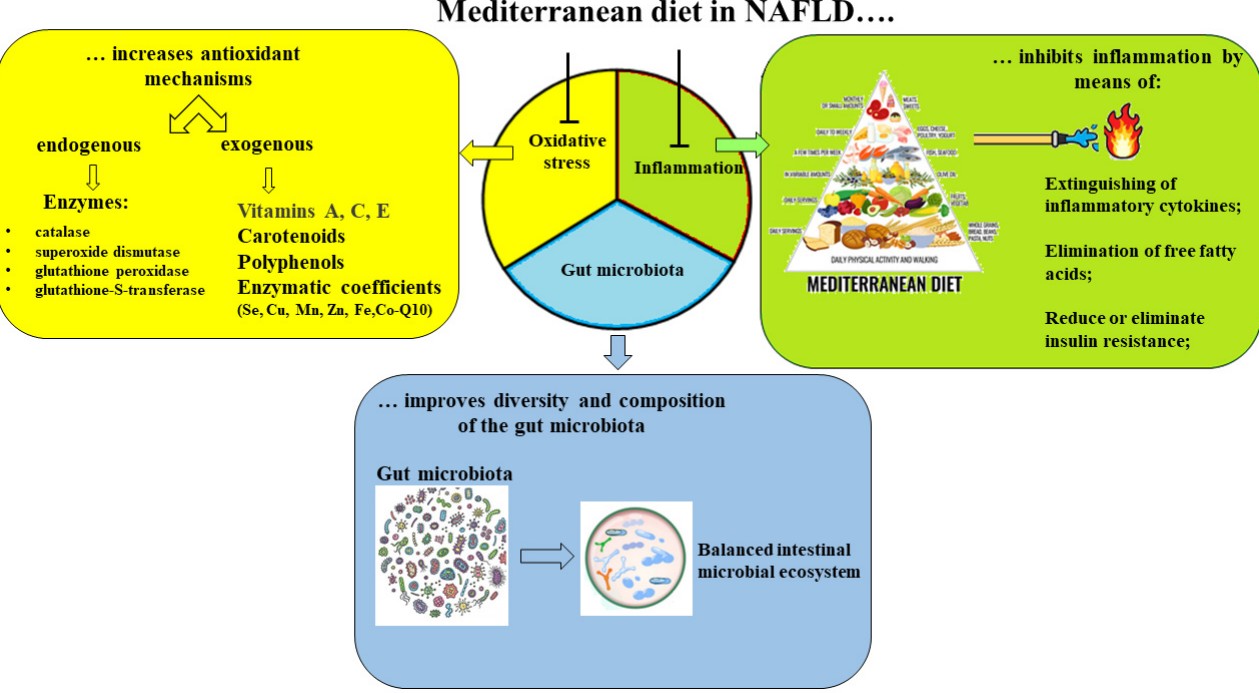

**Figure 2.** The Mediterranean diet protects from NAFLD. Three key mechanisms are used by the Mediterranean diet to protect against NAFLD: (1) increase antioxidant defenses to combat oxidative stress (portion colored in yellow); (2) inhibit the inflammatory process (portion colored in green); (3) improve the diversity and composition of the intestinal microbiota (portion colored in blue).

## 2. Phytochemical Properties of Bergamot Polyphenolic Fraction, *Cynara cardunculus* and Their Combination Product (Bergacyn®)

Among the innumerable foods included in the Mediterranean diet, we will examine two in particular: the bergamot fruit and the artichoke vegetable. The reasons for this choice are threefold: (1) both have a composition that justifies their antioxidant activity; (2) both are particularly widespread in the countries bordering the Mediterranean because of their climatic characteristics. (3) both are typical crops of the south of Italy and particularly of Calabria. The specification sheet and the comparison between BPF and *Cynara cardunculus* extract are shown in Figure 3. In particular, the tables show the description, the organoleptic properties, chemical characteristics and active ingredients.

**BPF: Bergamot Polyphenolic Fraction-Specification Sheet**

| DESCRIPTION | SPECIFICATIONS | METHODS |
|---|---|---|
| Botanical Source | *Citrus bergamia* Risso et Poit | |
| Family | Rutaceae | |
| Country of Origin | Calabria, Italy | |
| Part Used | Fruit | |
| Shelf Life | 3 years, if correctly stored | |
| ORGANOLEPTIC PROPERTIES | | |
| Colour | Yellow Powder | Visual (CQ-MO-148) |
| Odour | Aromatic | Visual (CQ-MO-418) |
| Flavour | Characteristic of bergamot | Visual (CQ-MO-418) |
| CHEM. CHARACTERISTICS | | |
| pH | 3.0-4.0 | IM (0.5% in water) at 25°C |
| Soluble in 50% H$_2$O+ EtOH | Good | Visual (CQ-MO-148) |
| Active Ingredient Strength | HPLC | |
| Pesticides Residue | Negative | PT CHIM 69rev02 011 |
| ACTIVE INGREDIENTS | UNIT | RANGE |
| Polyphenols (Neoeriocitrin, Naringin, Neohesperidin, Melitidin, Bruteridin) | % | 38% |

**Cynara cardunculus Extract-Specification Sheet**

| DESCRIPTION | SPECIFICATIONS | METHODS |
|---|---|---|
| Botanical Source | *Cynara cardunculus* | |
| Family | Asteraceae | |
| Country of Origin | Calabria, Italy | |
| Part Used | Leaf | |
| ORGANOLEPTIC PROPERTIES | | |
| Colour | Gold Green Powder | Visual (CQ-MO-148) |
| Odour | Aromatic | Visual (CQ-MO-148) |
| Flavour | Characteristic of Cynara leaf | Visual (CQ-MO-148) |
| CHEM CHARACTERISTICS | | |
| pH | 3.0-5.0 | IM (0.5% in water) at 25°C |
| Soluble in 50% H$_2$O+EtOH | Good | Visual (CQ-MO-148) |
| Active Ingredients Strengh | HPLC | |
| Pesticide Residue | Negative | PT CHIM 69rev02 011 |
| COMPONENT | UNIT | RANGE |
| Cynaroprin | % | 10 % |
| Total flavonoids (as luteolin-7-O-glucoside) | % | 15 % |
| Caffeolylquinic acis (as | % | 6 % |

**Figure 3.** Datasheets of BPF and *Cynara cardunculus*. The specification sheet and the comparison between BPF and *Cynara cardunculus* and chemical characteristics and the composition of the main components of both compounds. This data was provided by by HEAD srl (Bianco, Italy).

Interest in natural products has increased significantly, due to their beneficial effects on human health; one of the most promising is definitely bergamot (*Citrus bergamia*, Risso et Poiteau), a citrus fruit belonging to the *Rutaceae* family and to the genus *Citrus*, which grows in southern Italy, expressing the best quality in Reggio Calabria province, in Calabria region, Italy [52,53]. The reasons why the best specimens of bergamot prefer this geographical area are to be found in the climate and soil composition that fully meet the needs of the plant [54]. The bergamot is defined as a hybrid of bitter orange and lemon and it appears fleshy: it is similar to a modified berry, called hesperidium, and like the fruits of all citrus fruits belongs to the genus *Citrus* (lemon, orange, bergamot, cedar, etc.). The color of bergamot changes from yellow to green depending on the degree of ripeness and the taste of its juice is less sour than lemon, but more bitter than grapefruit [55]. There is still confusion and uncertainty about the botanical and geographical origins of this plant: it is possible that the bergamot is native to Calabria (Italy), as a result of mutations of the species; or it could come from Greece, the Antilles or from the Canary Islands, or have been imported by Christopher Columbus [56]. The historical use of Bergamot was concentrated in the perfumery, cosmetic, food and confectionery industries because of the intense fragrance of the fruit; however, folk medicine identified it as the main remedy against microbial infections, sores, fevers, mouth and skin infections, and respiratory system and urinary tract infections [57,58]. The bergamot, as well as the other citrus fruits, is mainly rich in flavonoids and has proved to possess beneficial properties in human health with antioxidant and anti-inflammatory properties [59,60], but also against immune disorders, high cholesterol, heart failure and coronary heart diseases [61,62]. A product derived from the fruit bergamot (both juice and albedo) is the polyphenolic fraction of bergamot (BPF) that is enriched with polyphenols: the main components are naringin, neohesperidin, neoeriocitrin and glycosylated polyphenols, such as melitidin and bruteridin [63]. The concentrated aqueous solution possesses a total concentration of polyphenols of 40%. Recent data have shown that bergamot polyphenols are able to exert

not only an antioxidant response both in vitro and in vivo [64–67], but also a reduction in the levels of glucose, cholesterol, serum triglycerides, systemic inflammation and an improvement in endothelial function [68–73].

Another plant product that has affected the scientific world for its beneficial properties on human health is the artichoke. The artichoke (*Cynara cardunculus* L.) belongs to the family of the *Asteraceae*, genus *Cynara*. This perennial plant is characterized by an annual growth cycle and includes three taxa: *Cynara cardunculus* L. subsp. *scolymus*; *Cynara cardunculus* L. var. *altilis*; *Cynara cardunculus* L. var. *sylvestris* [74,75]. This plant is mainly cultivated for the production of artichokes, which has a large fleshy head or capital and constitutes 30–40% of the fresh weight. The edible parts of the Artichoke are the tender inner leaves (bracts) and the container commonly known as "heart" [76]. The description of the genus *Cynara* dates back to the 4th Century B.C. in which the greek writer Theophrastus refers to these plants for food and medical purposes [77]. In the time of the Roman Empire, artichoke was used as a delicious appetizer and an effective digestive. This niche remained until the sixteenth century, when the use of this plant was for medicinal purposes to relieve liver fatigue. In traditional European medicine, artichoke leaves have been used as a stimulant for liver bile flow and as a diuretic [78]. To date, artichoke has used to improve liver function and to reduce serum cholesterol levels [79]. The plant grows preferably in arid regions characterized by warm and dry climate and, for this reason, the geographical area of the Mediterranean is very suitable [80]. To date, it is known that artichoke can also be used to obtain nutraceuticals and for the extraction of food additives [81]. Italian artichoke production exceeds that of other countries in the world and is approaching 474,000 tons per year, although its cultivation is present also in Greece, Spain, France, United States (mainly in California), South America (Argentina, Chile, Peru), North Africa, Turkey, Iran and China [82]. This expansion is facilitated by the genetic characteristics of the plant that characterize it as robust and tolerant to pathogens. The reason is to be found in its composition and mainly in sesquiterpenes, of which the best known is cynaropicrin [83]. *Cynara cardunculus* is a food that is part of our common diet and is particularly recommended in the Mediterranean diet. Phytochemical studies have shown that *Cynara cardunculus* extract is rich in caffeic acid derivatives (dicaffeoylquinic acid and mono-caffeoylquinic acid such as chlorogenic acid and cynarin), flavonoids (including luteolin-7-beta-glucoside, the glycosides luteolin-7-beta-rutinoside and luteolin-4-beta-D-glucoside), and sesquiterpenes such as 5–10% cynaropicrin [84]. The leaves of the Artichoke have been considered as active ingredients of this plant and used alone or in association with other herbs to prepare herbal teas or medicinal herbs. The chemical composition of artichoke leaves indicates a high content of polyphenols, flavones, fibers, vitamin C, minerals (phosphorus, sodium and potassium), inulin, and hydroxycinnamate [85]. The artichoke leaves have shown beneficial properties including antioxidative, hepato-protective, antibacterial and bile-expelling properties, and the ability to inhibit cholesterol biosynthesis and LDL oxidation [86]. In addition, in-depth studies have shown that artichoke leaves exert hypoglycemic, lowering cholesterol, choleretic, digestive, anti-atherosclerotic, antioxidant, cardiovascular, genotoxicity, anticancer, and prebiotic/probiotic effects [87,88]. Therefore, in the light of the listed beneficial properties of these plant extracts, the aim of the present review is to summarize the current evidence on the effects of BPF and *Cynara cardunculus* extract in NAFLD.

Finally, a formulation containing both extracts, an innovative and unique combination of BPF plus *Cynara cardunculus* extract, was obtained, mixing 150 mg of BPF powder with 150 mg of *Cynara cardunculus* extract. The mixture was encapsulated in capsules containing 300 mg of excipients represented by micronised bergamot pulp fibres. This formulation was identified as Bergacyn® and contained 20% polyphenols and 5% cinaropicrin. To date, studies have shown beneficial effects of Bergacyn® in reducing inflammation of the liver steatosis and oxidative stress in type 2 diabetes [89]. In addition, Bergacyn® is the first solvent-free extract, so it also demonstrates a higher safety profile than other products. This mixture showed a synergistic effect of the two components compared to the results

obtained by BPF or *Cynara cardunculus* extract alone. Further information on Bergacyn is given in Figure 4.

**Bergacyn-Specification Sheet**

| DESCRIPTION | |
|---|---|
| Trade name | Bergacyn |
| Botanical Source | *Citrus bergamia* Risso et Poit, and *Cynara cardunculus* |
| Family | Rutaceae (bergamot) and Asteraceae (artichoke) |
| Country of origin | Calabria, Italy |
| Part used | Fruit (bergamot) and leaf (artichoke) |
| **ORGANOLEPTIC PROPERTIES** | |
| Colour | Gold green Powder |
| Odour | Aromatic |
| Flavour | Characteristic of bergamot |
| **CHEM. CHARACTERISTICS** | |
| pH | 3.0-5.0 |
| Solubility in 40°C $H_2O$ | |
| Solubility in 50% $H_2O$ + EtOH | |
| Pesticide Residues | Negative |
| **ACTIVE INGREDIENTS** | **RANGE** |
| Bergamot Polyphenols (Neoeriocitrin, naringin, Nehesperidin, Melitidin, Brutieridin) | 19% |
| *Cynara cardunculus* (Cynaropicrin) | 5% |

**Figure 4.** Datasheet of Bergacyn. This table shows the description, the organoleptic and chemical characteristics and the composition of Bergacyn. This data was provided by HEAD srl (Bianco, Italy).

### 3. BPF and NAFLD

Since about 10 years, bergamot fruit has been selected because it has shown marked hypolipidemic effects in patients with dyslipidemic and cardiometabolic pathologies, which showed adverse effects to the drugs commonly used. In fact, this natural product has found a good effectiveness, a robust safety profile without any damage. For this reason, bergamot has proven to be an excellent candidate as a potential remedy to reduce cardiometabolic risk [90]. An elegant and very well-structured article by Mollace et al., showed that treatment with bergamot (1500 mg/day for 30 days and after a washing period of 60 days) was able to exert a dose-dependent lipid reduction effect [91]. In this study, 237 patients with hypercholesterolemia, hyperlipidemia or metabolic syndrome were enrolled, each of whom who had previously developed side effects when treated with statins. The hypothesized components responsible for the hypolipidemic effect of bergamot were polyphenols that—in addition to offer potential effectiveness against many pathological conditions such as diseases resulting from oxidative stress, chronic inflammation, cardiovascular diseases and metabolic diseases — exert without any reasonable doubt an hypolipidemic action [92]. To date, it is known that polyphenols obtained from bergamot juice and albedo (such as naringin, neoeriocyton, neohesperidin, melitidine and brutieridine) are the main proponents of the lipid-lowering effect: in fact, it has been shown that BPF, the bergamot fraction enriched with a polyphenolic content of 40%, is able to effect a reduction in lipid content greater than common bergamot juice [93,94]. The mechanisms of action involved in the lipid-lowering properties of bergamot have been extensively explored and can be summed up in the total excretion of bile acids, the increase of fecal neutral sterols [69,72,95], the statin-like action, due to the presence of melitidin and brutieridin, which are structural analogues of statins [96,97] and the reduction of cholesterol absorption, due to inhibition of pancreatic cholesterol ester hydrolase (pCEH) [70]. Mirarchi et al., have shown that BPF

reduces hepatocyte intracellular neutral lipids and triglycerides through increased beta-oxidation process. These interesting results were obtained in rat hepatic cells and replicated in human cells, using 3D spheroids composed of immortalized human hepatic and hepatic stellate cells [98]. Bergamot, and even more polyphenol enriched fraction (BPF), have been shown to act positively on NAFLD: in fact, Parafati et al. have shown in vivo that BPF can accelerate the therapeutic effects of weight loss induced by a standard normocaloric diet compared to harm reported in adults and children undergoing a dietary model based on "junk food" and named the "cafeteria diet" [99]. An unhealthy diet, as already mentioned, is an important contribution to many chronic diseases, such as NAFLD, obesity and hyperlipidemia. In the United States, 49% of children and 32% of adults feed on unhealthy diets, negatively affecting food preferences and health over their lifetime [100,101]. Junk foods include salty packaged snacks, sweet, sugary drinks and candies, from industrially processed foods, without healthy energy potential and which only increase caloric intake [102]. In the experimental model of Parafati et al. young rats, exposed to the cafeteria diet for 16 weeks, became obese early and developed NAFLD, with proven inflammatory characteristics. The group treated with BPF has reduced the triglycerides in the blood, the amount of liver fat by 90% and hepatic inflammation by decreasing the expression of interleukin 6, 10 and the inflammatory foci compared to the control group. In addition, the clearance of hepatic lipid droplets has been accelerated [99]. Since a large number of NAFLD models have several limitations, there was a need to create animal models that could perfectly reproduce the physiology and histology observed in humans with NAFLD. A preclinical model for NAFLD includes caloric excess, development of obesity, insulin resistance and dyslipidemia. This animal model of NAFLD is called "DIAMOND" in which animals are fed a diet very rich in fat and ad libitum water consumption with a high content of fructose and glucose (this condition is comparable to Western diets). The result is the induction of steatosis, liver damage and fibrosis in the animals involved [103]. Musolino et al. used the DIAMOND model, in which mice 8–12 weeks of age were fed a normal diet and tap water or a high fat/carbohydrate diet and a high fructose-glucose water solution for up to 27 weeks. After inducing liver damage and NAFLD, it was assessed whether pre-treatment with BPF could revert the effects. The results obtained showed that BPF protected animals with NAFLD, improving the typical pathophysiological characteristics; in particular, BPF reduced body weight, histological lesions, expression of liver enzymes, fibrosis, dyslipidemia and markers of oxidative stress [104]. The protective effect exerted by BPF on NAFLD patients was also demonstrated by Fogacci et al. who tested the supplementation of phytosome BPF, in association with other compounds (Artichoke extract, Coenzyme Q10 phytosome and zinc), in 60 patients. The mixture, called Eufortyn Cholesterol Plus, resulted in the reduction of systemic inflammation, serum lipids concentration (total cholesterol, low-density lipoprotein cholesterol, non-high-density lipoprotein cholesterol), indexes of NAFLD, C-reactive protein and the improvement in endothelial reactivity [105]. The use of natural products, to counteract an increase in triglycerides and serum cholesterol levels (factors that increase the risk of developing myocardial infarction, stroke, vascular lesions, hyperlipidemia, diabetes, metabolic syndrome and NAFLD), is increased exponentially thanks to the proven protective effect, free of the side effects developed with the use of the main drugs for the treatment of hyperlipidemias. Mollace et al. presented a very well articulated study, in which the protective effects of BPF were compared to those induced by red yeast rice in fed animals at hyperlipidemic diet. The animals took BPF or red yeast rice orally for 30 consecutive days. The results showed a lipid-lowering effect greater than BPF compared to red yeast rice: in fact, BPF was able to counteract the increase of serum LDL cholesterol and triglycerides with the concomitant reduction of biomarkers of oxidative stress, malondialdehyde and glutathione peroxidase serum levels. In contrast, red yeast rice caused only a poor reduction in serum LDL cholesterol [106]. In view of the above, much additional work is needed to determine whether BPF can also be used in humans for the treatment of NAFLD with the same efficacy and low side effects.

### 3.1. Cynara cardunculus Extract and NAFLD

The Artichoke, pre-standingly *Cynara cardunculus* L., is a plant food fundamental component of a traditional Mediterranean diet: in addition to its already mentioned properties, has potential lipid lowering and liver protection properties. The effect of artichoke leaf extract supplementation on the lipid pattern of 92 overweight subjects with hypercholesterolaemia has been evaluated. The supplementation showed a robust protection compared to the control group, so much to conclude that artichoke leaf extract could play a significant role in the management of hypercholesterolemia, promoting the increase of HDL and reducing total cholesterol and LDL cholesterol [107]. Numerous phytochemical studies have also shown that *Cynara cardunculus* extract is rich in antioxidants (such as dicaffeoylquinic acid, mono-caffeoylquinic acid, chlorogenic acid and cynarin), flavonoids (including luteolin-4-beta-D-glucoside, luteolin-7-beta-rutinoside and luteolin-7-beta-glucoside) and sesquiterpenes such as 5–10% cynaropicrin [108]. For this reason, *Cynara cardunculus* extract can be a valuable tool to combat hyperlipidemia and NAFLD. Most likely the mechanism of action must be sought in luteolin that is able to induce a hypolipemic effect by inhibiting the enzymes hydroxy-methyl-glutaryl-coenzyme A reductase and acetyl-coa C-acetyltransferase, binding to liver sterol regulatory element and increasing the excretion of fecal sterols [109,110]. Oppedisano et al. showed variations among rats fed a high-fat diet or a normal fat diet for 4 consecutive weeks. The first group showed a significant increase in serum glucose, cholesterol and triglycerides, an increase in body weight and histopathological characteristics compatible with NAFLD compared to rats fed a normal fat diet. The induced metabolic changes and hepatic steatosis have been antagonized by co-treatment with *Cynara cardunculus* extract 10 and 20 mg/kg [111]. In addition, artichoke supplementation has been shown to reduce the expression of liver enzymes in NAFLD patients, which is essential for expressing liver health and detoxifying capacity. Joint analyses, of eight clinical studies, have found that artichoke supplementation significantly reduces the concentration of aspartate aminotransferase and alanine transaminase in comparison with the control group [112]. At the same time, another study examined the protective effects of artichoke leaf extract on the liver of NAFLD patients: in this study, subjects suffering from NAFLD were treated with 600 mg/day of artichoke leaf extract for a period of two months. Patients who were given artichoke extract showed reduction of total cholesterol, low-density lipoprotein cholesterol, triglyceride concentrations, liver enzymes and total bilirubin compared to the control group [113]. Lee et al. evaluated the effect of artichoke leaf extract in two experimental models: in vivo on the livers of mice with NAFLD induced as a result of high fat/high fructose diet; in vitro on HepG2 liver cells in which oxidative stress was induced following treatment with $H_2O_2$. In the animal model, the administration of artichoke leaf extract determined the reduction of serum lipids, gamma-glutamine transferase, aspartate transaminase, alanine aminotransferase, bilirubin, mRNA levels of proinflammatory cytokines and apoptosis. In liver cells, the administration of the plant product suppressed inflammation and apoptosis, both caused by oxidative stress induced by hydrogen peroxide. The results obtained suggest, therefore, the integration with artichoke leaf extract to eliminate the suffering of hepatocytes during the development of NAFLD and to prevent the progression of hepatic steatohepatitis and non-alcoholic steatohepatitis [114]. Artichoke leaf extract has also been tested together with other compounds, in order to increase its effectiveness.

### 3.2. Synergistic Effect of Combination Product (Bergacyn®) in NAFLD

In order to evaluate the effects of *Cynara cardunculus* extract together with other drugs, we mention an important study of Majnooni et al. in which the effect of co-administration of artichoke leaf extract together with conventional drugs, on patients with NAFLD, was investigated. In particular, the difference of three treatments on this pathology has been evaluated: (1) metformin-vitamin E; (2) metformin-artichoke leaf extract; (3) vitamin E-artichoke leaf extract. The effect of the three treatments was compared after 12 weeks through the study of biochemical markers and liver ultrasonography. The results showed

that the greatest reduction in liver enzyme and liver fat values was found with treatments 2 and 3, highlighting that the concomitant use of artichoke leaf extract with both metformin and vitamin E can improve the basal state of NAFLD and its complications [115]. Subsequently, the research of the synergistic effects of artichoke leaf extract with other compounds led us to the study of *Cynara cardunculus* extract + BPF. For this reason, the effect of Bergacyn® (the mixture of these plant extracts) has been evaluated: NAFLD is often worsened by the concomitant presence of type 2 diabetes mellitus, leading to aggravation of both inflammatory and fibrotic processes. In a clinical study, 80 adult patients with a history of type 2 diabetes and NAFLD were treated with BPF (300 mg/daily) together with *Cynara cardunculus* extract (300 mg/daily) taken separately or formulated in combination 50/50% (Bergacyn®; 300 mg/daily). In parallel, a control group took placebo. The results obtained showed a significant improvement of NAFLD biomarkers in diabetic patients. The mechanism of action seems to involve the reduction of oxidative stress, inflammation and Nitric oxide-mediated vasodilation. In addition, the effect of Bergacyn® has been synergistic, suggesting the use of this formulation to counteract endothelial dysfunction and vascular inflammation in patients suffering from NAFLD and type 2 diabetes [116]. Another recent and interesting study was carried out on 102 patients with NAFLD who showed a significant increase in liver fat compared to the control group. These subjects were treated with a nutraceutical containing BPF together with *Cynara cardunculus* extract for 12 weeks. In contrast, the control group took placebo daily. Liver fat content, serum transaminases, lipids and glucose were measured at the beginning and end of the study. The results obtained showed a significant reduction in liver fat content in women over 50 years of age, and of hepatic steatosis compared to the control group [117].

### *3.3. Effect of Bergacyn® in Vascular Consequences of NAFLD*

Since NAFLD can be found in 65–70% of patients with type 2 diabetes mellitus (T2DM), this liver disease could pose an additional cardio-metabolic risk that could worsen the pro-inflammatory and pro-fibrotic profile [118,119]. We could say that NAFLD is a predisposing factor for the development of T2DM and its complications. In fact, NAFLD is associated with an early impairment of fasting glucose and insulin resistance. Additionally, patients with T2DM develop steato-hepatitis easily [120,121]. Moreover, T2DM sufferers have been shown to worsen the condition of pre-existing or concomitant NAFLD with the acceleration of symptoms of di-steato-hepatitis, hepatic fibrosis and hepato-carcinoma [122]. NAFLD is also represented by secondary events including vascular endothelial dysfunction, which is responsible for altering the maintenance of vascular tone, metabolite transport, haemostasis, inflammation, thrombosis and angiogenesis [123]. Treatment with Bergacyn® showed a significant improvement in these dysfunctions allegedly mediated by induced NO release from endothelial cells, which is generally compromised in both NAFLD and T2DM [124,125]. Bergacyn® seems to contribute significantly to the resolution of the above dysfunctions and the two components of which composed act synergistically. In fact, the co-milling and micronization of both nutraceuticals leads to a better tissue absorption and distribution of administered compounds than the single doses of both products [126]. Although the mechanism of synergy of the components of Bergacyn® must be better clarified, it is conceivable that the polyphenols contained and sesquiterpenes may be responsible for reducing oxidative stress, anti-inflammatory responses and improved liver function [127]. Finally, it is also likely to think that an improved fat metabolism, Bergacyn® induced, is able to ensure the bioavailability of arginine and, consequently, put into operation the machine that generates NO [128,129].

### 4. Discussion and Conclusions

The multitude of data reported in this review highlighted the protective role of BPF and *Cynara cardunculus* extract in dyslipidemic diseases, with particular reference to NAFLD, in accordance with the growing interest in the use of plant nutraceuticals as potential treatment strategies against numerous pathologies [130]. To date, it is known that herbal

bioactive compounds contribute in the maintenance of health, promoting longevity and a better quality of life, so much so that the global interest in nutraceuticals is enormous and, in the United States, generates an economic income of billions of dollars [131]. Dyslipidemic conditions, such as obesity, metabolic syndrome, type 2 diabetes, NAFLD, are characterised, among others, by moderate risk factors linked to lifestyle, the chosen dietary model and physical activity [132]. In fact, an energy-balanced diet low in saturated fat (which should not exceed 7% of the total energy), a reduction in body weight (5–10%) for obese or overweight individuals, a physical activity intensity (about 150 min/week), avoidance of active or passive exposure to tobacco smoke and restriction of alcohol, simple sugars, and refined carbohydrate intakes can all help reduce low-density lipoprotein cholesterol, triglyceride levels and ectopic fat, which underlie the development of dyslipidemias [133]. To date, it is known that the dietary factor is the most important among those listed and have been identified and clinically tested many nutraceuticals with effectiveness and tolerability [134]. Among them, it is worth mentioning bergamot polyphenol fraction, artichoke, plant sterols, red yeast rice, green tea, garlic, soluble fibers, berberine and spirulina. Lipid-lowering nutraceuticals can be classified according to their mechanism of action and divided into three classes: (1) inhibitors of intestinal absorption of cholesterol, (2) inhibitors of hepatic synthesis of cholesterol, and (3) LDL cholesterol excretion enhancers [135]. The first mechanism of action includes the reduction of intestinal absorption of exogenous cholesterol micelles in the gastrointestinal lumen, with the concomitant aid of increased gastric emptying time, increased satiety and faecal excretion of cholesterol and bile salts [136]. The second mechanism of action is based on the inhibition of the enzyme 3-hydroxy-3-methyl-glutaryl-coenzyme A reductase, necessary in the biosynthesis of cholesterol and target of the widely available cholesterol-lowering drugs collectively as the statins, which help treat dyslipidemia. In particular, BPF acts as a statin inhibiting this enzyme and reducing apoB lipoproteins and the formation of cholesterol esters. In addition, BPF increases the faecal excretion of cholesterol by reducing intestinal absorption and increasing the excretion of bile acids [137]. The lipid-lowering mechanisms of *Cynara cardunculus* extract and artichoke and are the interaction of luteolin with 3-hydroxy-3-methyl-glutaryl-coenzyme A reductase, and the regulation of pathways of proteins binding sterol regulatory elements in the liver [138]. Finally, the mechanism "LDL cholesterol excretion enhancers" is based on the clearance of cholesterol from the blood, the reduction of cholesterol biosynthesis and the increase in the fecal excretion of bile salts [139,140]. Statins are currently the drugs of choice for dyslipidemia, although they are high-risk drugs with side effects that often reduce adherence to therapy [141,142]. Since many scientific studies have demonstrated lipid lowering effects following treatment with BPF and *Cynara cardunculus* extract, when considering the treatment of NAFLD patients these nutraceutical remedies could be administered alone or in combination with statins in order to help patients achieve their goals and improve endothelial function with minimal side effects [143,144]. Some clinical studies have established that the use of many nutraceuticals, with hypolipidemic action, could allow the reduction of the dosages of statins, taken in concomitance, without reducing the results [145,146]. Alternatively, combinations of nutraceutical lipid lowering could be taken, reducing the dosages of individual components and improving their effectiveness [147,148]. This scientific field has yet to be fully explored and further clinical studies will be needed to establish effective and safe treatment against NAFLD. For example, the combination of BPF and *Cynara cardunculus* extract, two extremely interesting compounds in the treatment of NAFLD, took place synthetically in the laboratory and created Bergacyn. This formulation seems to be effective even if, at the moment, it has been characterized exclusively for the content of bergamot polyphenols and artichoke cynaropicrin. For this reason, it would be essential to carry out more in-depth studies that can justify its use in vivo and in clinical trials. Overall, the data in the literature provide three conclusive information:

(1) artichoke, in combination with other compounds with similar functions, can demonstrate a synergistic effect;

(2) 　artichoke and BPF can be used as a complementary therapy for the treatment of dyslipidemic diseases such as NAFLD;

(3) 　Bergacyn® can outperform the effects of Cynara Cardunculus extract and BPF taken separately. Bergacyn® is patented for composition, manufacturing process and application in the therapeutic support of healthy liver fat levels [89].

**Author Contributions:** J.M. and V.M. (Vincenzo Mollace) conceptualized and designed the manuscript; J.M., F.O. and R.M. (Rocco Mollace), wrote the manuscript; F.B., F.S., S.N., S.R. and L.G. have revised revising critically the manuscript; R.M. (Roberta Macrì), I.B., C.C., M.G. and V.M. (Vincenzo Musolino), A.C., A.R.C., A.B., V.S., D.S. and E.P. participated in drafting the article. All authors have read and agreed to the published version of the manuscript.

**Funding:** The work was supported by public resources from the Italian Ministry of Research.

**Institutional Review Board Statement:** Not available.

**Data Availability Statement:** Not available.

**Acknowledgments:** This work was supported by PONa3 00359.

**Conflicts of Interest:** The authors declare no conflict of interest.

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
