# Peer review of "The Phytochemical Synergistic Properties of Combination of Bergamot Polyphenolic Fraction and Cynara cardunculus Extract in Non-Alcoholic Fatty Liver Disease"

_agriculture, doi:10.3390/agriculture13020249_

Round 1

Reviewer 1 Report

Non-alcoholic fatty liver disease (NAFLD) is one of the leading causes of liver related morbidity and mortality. The authors investigated the effects of the polyphenolic fraction of bergamot, Cynara Cardunculus extract and a combination of these two products. The paper contains interesting information. However, I have some suggestions:

1. Authors mentioned balanced diet is a keyway to treat NAFLD, however, are bergamot and Cynara Cardunculus the most important diet? Authors should add at least one paragraph to elucidate this. Why chooses these plants?

2. In Section Phytochemical properties, I suggest adding a table to summary the phytochemistry of the two plants, and number all the compounds (polyphenolic compounds) isolated or identified from Bergamot and Cynara cardunculus.

3. Is there any difference between phytochemistry of combination product and Bergamot or Cynara cardunculus alone. Some new compounds produced after the combination? The change of phytochemistry after combination should be added.

4. Authors should focus on the synergistic effect, which compounds present the synergistic effect, how the synergistic effect works? The Discussion should keep to the point (synergistic effect).

Author Response

Dear Reviewer,
thanks for your valuable suggestions. The manuscript has been revised and I hope it will be considered suitable for publication.

Jessica

Reviewer 2 Report

In general the article is good, some punctuation marks are needed in order to clarify some ideas, such as in  the following lines:

127, a comma after "processes"

299-301, 302 -303: better organization of ideas  

Some misspelles or repeated words, like in:

line 300, "or hyperlipidem", it looks 

line 272: a misspelled word: "amaunt"

line 343: an extra S in NAFLDS

The idea in 263-265 looks lost, 

Author Response

(The authors gave the same response as above.)

Reviewer 3 Report

The paper reviewed the synergistic effectiveness of combination of Bergamot Polyphenolic Fraction and Cynara Cardunculus extract in NAFLD, suggesting a potentially  approch in preventing and treating NAFLD. Some questions should be further clarified. Are the synergistic effects only observed in 50/50% of Bergamot Polyphenolic Fraction and Cynara Cardunculus extract? More evidences including in rodents should be listed and overviewed.

Author Response

(The authors gave the same response as above.)

Round 2

Reviewer 1 Report

Thanks for the revision. It is much better. However, the Latin name of plants should be written in italic, and the initial of specific name should be written in lower case, such as L558, L221, L224, and Figure 3. Please check this through the paper.

Author Response

Dear reviewer,
thank you for your valuable tips and suggestions. 

As you indicated, we have corrected the Latin name of plants in both text and Figures 3 and 4.

Jessica